# Comprehensive Genomic Profiling Reveals Clinical Associations in Response to Immune Therapy in Head and Neck Cancer

**DOI:** 10.3390/cancers14143476

**Published:** 2022-07-18

**Authors:** Rika Noji, Kohki Tohyama, Takuma Kugimoto, Takeshi Kuroshima, Hideaki Hirai, Hirofumi Tomioka, Yasuyuki Michi, Akihisa Tasaki, Kazuchika Ohno, Yosuke Ariizumi, Iichiroh Onishi, Mitsukuni Suenaga, Takehiko Mori, Ryuichi Okamoto, Ryoichi Yoshimura, Masahiko Miura, Takahiro Asakage, Satoshi Miyake, Sadakatsu Ikeda, Hiroyuki Harada, Yoshihito Kano

**Affiliations:** 1Department of Clinical Oncology, Graduate School of Medical and Dental Sciences, Tokyo Medical and Dental University (TMDU), 1-5-45 Yushima, Bunkyo-Ku, Tokyo 113-8510, Japan; nojiri1207.osur@tmd.ac.jp (R.N.); suenaga.srg2@tmd.ac.jp (M.S.); mori.hema@tmd.ac.jp (T.M.); sm.conc@tmd.ac.jp (S.M.); ikeda.canc@tmd.ac.jp (S.I.); 2Department of Oral and Maxillofacial Surgery, Division of Health Science, Graduate School of Medical and Dental Sciences, Tokyo Medical and Dental University (TMDU), 1-5-45 Yushima, Bunkyo-Ku, Tokyo 113-8510, Japan; kugimoto.osur@tmd.ac.jp (T.K.); kuroosur@tmd.ac.jp (T.K.); hirai.osur@tmd.ac.jp (H.H.); tomy.osur@tmd.ac.jp (H.T.); y-mic.mfs@tmd.ac.jp (Y.M.); hiro-harada.osur@tmd.ac.jp (H.H.); 3Department of Precision Cancer Medicine, Center for Innovative Cancer Treatment, Tokyo Medical and Dental University (TMDU), 1-5-45 Yushima, Bunkyo-Ku, Tokyo 113-8510, Japan; 4Department of Oral Radiation Oncology, Division of Oral Health Science, Graduate School of Medical and Dental Sciences, Tokyo Medical and Dental University (TMDU), 1-5-45 Yushima, Bunkyo-Ku, Tokyo 113-8510, Japan; kokimdth@tmd.ac.jp (K.T.); masa.mdth@tmd.ac.jp (M.M.); 5Department of Head and Neck Surgery, Tokyo Medical and Dental University (TMDU), 1-5-45 Yushima, Bunkyo-Ku, Tokyo 113-8510, Japan; tasoto@tmd.ac.jp (A.T.); ohno.hns@tmd.ac.jp (K.O.); ariizumi.hns@tmd.ac.jp (Y.A.); tasakage.hns@tmd.ac.jp (T.A.); 6Department of Pathology, Graduate School of Medical and Dental Sciences, Tokyo Medical and Dental University (TMDU), 1-5-45 Yushima, Bunkyo-Ku, Tokyo 113-8510, Japan; iichpth2@tmd.ac.jp; 7Department of Hematology, Tokyo Medical and Dental University (TMDU), 1-5-45 Yushima, Bunkyo-Ku, Tokyo 113-8510, Japan; 8Department of Gastroenterology and Hepatology, Tokyo Medical and Dental University (TMDU), 1-5-45 Yushima, Bunkyo-Ku, Tokyo 113-8510, Japan; rokamoto.gast@tmd.ac.jp; 9Department of Radiation Therapeutics and Oncology, Graduate School of Medical and Dental Sciences, Tokyo Medical and Dental University (TMDU), 1-5-45 Yushima, Bunkyo-Ku, Tokyo 113-8510, Japan; ysmrmrad@tmd.ac.jp

**Keywords:** head and neck cancer (HNC), comprehensive genomic profiling (CGP), immune checkpoint inhibitor (ICI), biomarker, tumor mutational burden (TMB), Center for Cancer Genomics and Advanced Therapeutics (C-CAT)

## Abstract

**Simple Summary:**

The response rate of head and neck cancer (HNC) to immune checkpoint inhibitors (ICIs) can be as low as 20%. Biomarkers need to be elucidated to predict therapeutic response. Comprehensive genomic profiling (CGP) provides information on cancer-related genetic aberrations. Among the 1100 HNC cases in the Center for Cancer Genomics and Advanced Therapeutics (C-CAT) database, only 5% received biomarker-matched therapy, including *NTRK* fusion. Cases of squamous cell carcinoma (SCC) with a high tumor mutational burden (TMB) (≥10 Mut/Mb) showed long-term survival (>2 years) in response to ICI therapy. However, the cases with CCND1 amplification showed a significantly lower response to ICI therapy. Therefore, *CCND1* amplification may serve as a negative prognostic marker. Therefore, CGP may be useful for establishing prognostic biomarkers for immunotherapy in patients with HNC.

**Abstract:**

Comprehensive genomic profiling (CGP) provides information regarding cancer-related genetic aberrations. However, its clinical utility in recurrent/metastatic head and neck cancer (R/M HNC) remains unknown. Additionally, predictive biomarkers for immune checkpoint inhibitors (ICIs) should be fully elucidated because of their low response rate. Here, we analyzed the clinical utility of CGP and identified predictive biomarkers that respond to ICIs in R/M HNC. We evaluated over 1100 cases of HNC using the nationwide genetic clinical database established by the Center for Cancer Genomics and Advanced Therapeutics (C-CAT) and 54 cases in an institution-based study. The C-CAT database revealed that 23% of the cases were candidates for clinical trials, and 5% received biomarker-matched therapy, including *NTRK* fusion. Our institution-based study showed that 9% of SCC cases and 25% of salivary gland cancer cases received targeted agents. In SCC cases, the tumor mutational burden (TMB) high (≥10 Mut/Mb) group showed long-term survival (>2 years) in response to ICI therapy, whereas the PD-L1 combined positive score showed no significant difference in progression-free survival. In multivariate analysis, *CCND1* amplification was associated with a lower response to ICIs. Our results indicate that CGP may be useful in identifying prognostic biomarkers for immunotherapy in patients with HNC.

## 1. Introduction

The head and neck area consist of several anatomical areas such as the oral cavity, pharynx, larynx, nasal passages, sinuses, thyroid glands, and salivary glands [1]. Most head and neck cancer (HNC) patients experience recurrence or metastasis despite aggressive multidisciplinary approaches, including surgical resection, radiation therapy, and adjuvant chemotherapy [2,3]. In recent years, immune checkpoint inhibitors (ICIs) have been developed and clinically applied to various cancers. Anti-programmed cell death-1 (PD-1) protein monoclonal antibodies nivolumab and pembrolizumab are approved for the treatment of patients with recurrent or metastatic head and neck cancers (R/M HNC) [1]. The binding of the PD-1 receptor to programmed cell death ligand-1 (PD-L1) results in the suppression of T cell immunological responses and serves as a mechanism of tumor immune evasion [4]. ICIs can block suppressive signaling through the PD-1/PD-L1 pathway and enhance antitumor immune activity [5]. However, the response rate has been reported to be 13–17% [3,6], and only a few cases showed a long-term response. Therefore, identification of predictors of ICI efficacy has been attempted for various cancers, although predictive markers have not been identified [7]. Currently, biomarkers, such as the combined positive score (CPS) and microsatellite instability (MSI), are used to select patients for clinical ICI therapy [8]. However, PD-L1 prediction results remain inconsistent, and MSI varies between cancer types; therefore, these biomarkers are limited [9]. An alternative biomarker is the tumor mutational burden (TMB) measured by next-generation sequencing (NGS). It has been suggested that a higher TMB is associated with higher response rates to ICIs in solid tumors. However, this association is still inconclusive, and no consensus has been reached regarding its widespread clinical use [10]. Additionally, a limited number of studies have shown a correlation between high TMB and response to ICIs in HNC [11]. There is an urgent need to identify effective biomarkers that respond to ICIs in R/M HNC. Recently, comprehensive genomic profiling (CGP) using NGS has been introduced in clinical practice for the selection of effective targeted therapies. Cancer treatment has become increasingly personalized based on the genomics of individual tumors [12]. However, because of the anatomical and pathological complications in HNC, there are only a limited number of reports on the clinical use of genomic data [13,14]. Therefore, it is necessary to analyze the relationship between the accumulated genomic data and the effects of therapeutic agents on HNC.

In this study, we investigated the database of national genomic and clinical information constructed by the Center for Cancer Genomics and Advanced Therapeutics (C-CAT), an organization established in the National Cancer Center, Japan, under the national policy to evaluate real-world data on genetic and clinical utility in R/M HNC. Moreover, we analyzed gene mutations in each histological type using the tumor profiling gene panel test in patients with R/M HNC and evaluated the achievement rate of targeted therapy for clinical utility. Next, in the subgroup analysis, we evaluated the association between ICI response in patients with head and neck squamous cell carcinoma (HNSCC) and multiple predictor candidates, including TMB and PD-L1 expression rates and gene alterations.

## 2. Materials and Methods

### 2.1. Patient Population on C-CAT Database

We queried an anonymized database of genomic and clinical information on patients with cancer, collected from core hospitals using C-CAT. The clinical data of C-CAT included age, sex, histology, treatment before and after the oncogene panel test, response to a drug, and type of CGP test. This study on R/M HNC patients was approved by the C-CAT Information Utilization Review Committee (proposal control number: CDU2022-021N).

### 2.2. Patient Population at Tokyo Medical and Dental University (TMDU)

We performed an oncogene panel test for R/M HNC patients who completed the standard therapy at our institution. Patients classified as having squamous cell carcinoma (SCC) or non-SCC were evaluated for the type of genetic alteration and achievement rate of targeted therapy. This study was approved by our institutional review board (approval number: G2020-020), and all patients provided written informed consent for using genomic and clinical data for research purposes.

### 2.3. Comprehensive Genomic Profiling (CGP) 

The C-CAT database included genomic information from the Foundation One^®^ Companion Diagnostic (F1CDx; Foundation Medicine, Inc., Cambridge, MA, USA) test, Foundation One^®^ Liquid Companion Diagnostic (F1LCDx; Foundation Medicine, Inc.) test, and National Cancer Center (NCC) Oncopanel test. F1CDx and F1LCDx detected substitutions, insertions, deletions, and copy number alterations in 324 genes and selective gene rearrangements and genomic signatures such as MSI and TMB [15]. The NCC Oncopanel test examines mutations, amplifications, and homozygous deletions of the entire coding region of 127 genes of clinical or preclinical relevance, along with rearrangements [16].

CGP was performed at our institution using the F1 test (F1CDx or F1LCDx). The F1CDx assay uses formalin-fixed paraffin-embedded (FFPE) tumor tissue samples obtained by biopsy or surgical operation, and pathologists select suitable tumor specimens for testing. One patient with no tissue sample underwent the F1LCDx test using blood samples. 

### 2.4. Molecular Tumor Board 

Each result in the CGP test report was discussed at a molecular tumor board meeting by a multidisciplinary team. The board included clinical oncologists, pathologists, clinical geneticists, bioinformatics researchers, genomic researchers, and genetic counselors. Board members discussed genetically informed treatment options and issues such as the interpretation of somatic/germline variants. Actionable mutations are defined as genomic alterations that satisfy the following conditions: gene alteration is characterized as a target of molecular targeted therapy. A drug is available for human use either as an antibody or as a small molecule compound with an IC50 concentration in the nanomolar range [17]. Evidence-level classification of gene aberrations was decided according to the Clinical Practice Guidance for NGS in Cancer Diagnosis and Treatment Edition 2.1 issued by the Japanese Society of Medical Oncology, Japan Society of Clinical Oncology, and Japanese Cancer Association [18]. 

### 2.5. Efficacy Assessments of ICI Therapy

Anti-PD-1 agents such as nivolumab or pembrolizumab were used. To evaluate the efficacy of ICIs administered alone, patients who received ICI combined with chemotherapy were excluded from response analysis. ICI therapy outcomes were evaluated based on the best overall response (OR) and progression-free survival (PFS) rates. Tumor response was evaluated according to the Response Evaluation Criteria in Solid Tumors (RECIST) 1.1 criteria [19]. The objective response rate (ORR) was defined as the number of patients who achieved complete response (CR) or partial response (PR). Disease control rate (DCR) was defined as the number of patients with CR, PR, or stable disease (SD). PFS was defined as the period from the start of ICI therapy to disease progression (DP). Patients lost to follow-up were censored on the last appointment date. PD-L1 protein expression was determined using CPS in FFPE tumor samples using the PD-L1 immunohistochemistry (IHC) 22C3 pharmDx assay (Agilent Technologies, Santa Clara, Calif., USA). The PD-L1 CPS was evaluated in three groups (CPS < 1, 1 ≤ CPS < 20, and CPS ≥ 20). TMB, defined as the total number of somatic mutations in a defined region of the tumor genome, was measured using the CGP test [20]. TMB was divided into three groups: low (5 Muts/Mb or less), intermediate (6–9 Muts/Mb), and high scores (10 Muts/Mb or more).

### 2.6. Statistical Analysis

Statistical analyses were performed using the EZR (Easy R) software package version 1.54 (Saitama Medical Center, Jichi Medical University, Saitama, Japan). PFS was calculated using the Kaplan–Meier method. Log-rank tests were used to determine significant differences between the two groups. Hazard ratios (HRs) were computed using a Cox proportional hazards model. Binary logistic regression analysis was performed to assess the clinical predictors of the response. All statistical tests were performed using a significance cut-off of *p* < 0.05.

## 3. Results

### 3.1. Patient Characteristics and Clinical Utility of CGP in C-CAT Data

From June 2019 to April 2022, among the 29,490 patients registered in C-CAT, 1119 were diagnosed with R/M HNC (Figure 1A). Of these, 1007 and 112 patients were analyzed using the F1 test and NCC Oncopanel test, respectively. The F1 test was performed in 369 SCC patients and 638 non-SCC patients. NCC Oncopanel test was performed in 48 patients with SCC and 64 patients with non-SCC. More than 70% of non-SCC cases were salivary gland tumors and carcinoma histological types, such as adenoid cystic, salivary duct, mucoepidermoid, and mammary analog secretory carcinoma. Of the 1119 patients with R/M HNC, 1050 (94%) had at least one detected genetic aberration, and 256 (23%) had an approved targeted agent or clinical trial candidate (Figure 1B). Fifty-one (5%) patients received targeted agents in accordance with gene aberrations. Among these, 16 patients had SCC, 31 had non-SCC, and four had unknown SCC. Non-SCC cases included patients who received biomarker-matched treatment for *NTRK3* fusion (14%) or *ERBB2* amplification (3%).

### 3.2. Patient Characteristics in TMDU

Between 27 December 2018 and 31 March 2022, 54 patients diagnosed with R/M HNC underwent CGP with the F1 test at TMDU. Of these patients, 34 had SCC, and 20 did not (Figure 2A, Table 1). The median age of the patients with SCC at the time of the F1 test was 60 years (range: 42–81 years), and 62% were male. For non-SCC patients, the median age was 61 years (range: 15–74 years), and 55% were female. The histological types of non-SCC were adenoid cystic carcinoma (35%), salivary duct carcinoma (15%), and carcinoma ex pleomorphic adenoma (10%). HPV positivity was 3–5% in all cases.

### 3.3. Sequencing Results

We evaluated the top 30 most frequent genetic mutations detected in 54 HNC patients with pharmacological intervention based on actionable genomic alterations at our institution (Figure 2B,C). Although some reports have shown that smoking history is associated with an increase in TMB [21,22], no significant correlation was observed. The percentages of frequent genes and variant types in SCC and non-SCC are shown in Figure 2D. The median TMB was 5 Muts/Mb (0–34) for SCC and 3 Muts/Mb (0–11) for non-SCC cases, which did not differ from the C-CAT data. The median TMB was 4 Muts/Mb (0–48) for SCC and 3 Muts/Mb (0–82) for non-SCC cases in C-CAT. The commonly mutated genes, including missense, nonsense, and splice site mutations, detected in SCC were *TP53* (62%), followed by the *TERT promoter* (56%), *CDKN2A* (21%), and *PIK3CA* (18%). The frequently amplified genes were *CCDN1*, *FGF3*, *FGF4*, *FGF19* (18%), *EGFR* (9%), and *MYC* (6%). In non-SCC cases, mutations in *TP53* (30%), deletions in *CDKN2A/B* (25%), and *MTAP* (20%) were found. Activating mutations or amplifications of *ERBB2* and *NTRK3* were found in 20% and 5% of cases, respectively, although *NTRK3* fusion was not observed. 

At least one genetic aberration, excluding the variance of uncertain significance (VUS), was detected in 33/34 SCC patients and 19/20 non-SCC patients (Figure 2C). Actionable mutations were discovered in 30/34 SCC patients and 15/20 non-SCC patients. According to their genetic aberrations, 9% of SCC cases and 25% of salivary gland cancer cases received targeted agents. Patients with SCC were subjected to immunotherapy for high TMB, whereas those with salivary gland cancers were treated with molecularly targeted drugs against *ERBB2* mutations.

### 3.4. Outcomes of ICI Therapy

First, we examined the PFS of 50 R/M HNSCC patients who received ICI monotherapy in our department (Figure 3A). The median PFS was 2.1 months (95% CI, 1.5–2.7), consistent with the previously reported median PFS of ICI in HNSCC [3]. All 34 patients with SCC who underwent panel testing in this study received ICI therapy (Table 2). Of these, one patient who received ICI combined with chemotherapy was excluded. One case was not evaluable (NE) because it was interrupted after the first session due to the infusion reaction. Six (18%) patients experienced PR, and six (18%) had SD; the ORR was 18%.

### 3.5. Association between TMB and ICI Response

ICI has been reported effective in high TMB cases [23,24]. We divided the patients into 3 groups according to TMB values to evaluate the differences in treatment efficacy. Of the 33 patients with SCC in TMDU, 21 (64%) were in the low TMB (≤5 Muts/Mb) group, 7 (21%) in the intermediate (6–9 Muts/Mb) group, and 4 (12%) were in the high (≥10 Muts/Mb) group (Figure 3B), which did not seem to differ from the C-CAT data [67% in low (≤5 Muts/Mb), 22% in intermediate (6–9 Muts/Mb) group, and 12% in high (≥10 Muts/Mb) TMB group]. TMB was not detected in 1 patient because of the small percentage of tumors in the specimen. The best response to ICI in the TMB-low group was PR in 1/21 (5%) and SD in 4/21 (19%) patients, and therefore, ORR was 5% (Table 2).

The TMB-high group showed PR in 2/4 (50%) patients, SD in 1/4 (33%) patients with an ORR of 50%, and DCR of 75%. In addition, the maximum percentage change in the tumor from baseline for each patient is displayed in a waterfall plot, and these time courses are displayed in a spider plot (Figure 3C). The low-TMB group (blue bars and lines) showed rapid growth within the first three months of treatment. In contrast, in the TMB-intermediate (yellow bars and lines) or high-TMB (red bars and lines) groups, many cases showed shrinkage or disease control. In particular, the two patients in the TMB-high group showed a long-term response of >24 months. 

Detailed data for the four cases in the TMB-high group are presented in Table 3. Two cases of oral cancer, namely, one of paranasal sinus cancer and one of cancer of unknown primary (CUP), were included. CUP could not be evaluated because of the infusion reaction. Case 1 had a postoperative recurrence of buccal mucosal carcinoma and multiple cervical lymph node metastases. Nivolumab was administered as first-line therapy, but the patient developed diarrhea (grade 3) after the first dose and it was discontinued. Diarrhea was resolved via steroid therapy, and the patient underwent the F1CDx test to identify potential therapeutic agents. The panel test results showed high TMB; therefore, we decided to rechallenge ICI with pembrolizumab in agreement with the patient. The patient showed marked tumor shrinkage of the primary and metastatic lesions since the start of treatment and long-term survival of >26 months without recurrence of enteritis. In case 2, a pleural tumor appeared after surgery for maxillary sinus cancer. A pathological review of the specimens revealed poorly differentiated sarcomatous tissue; however, no definite diagnosis was made. To explore a therapeutic strategy, panel testing was performed on both the head and neck tissues and pleural tumors. Panel testing showed that gene mutations, which are frequently mutated in HNSCC, were shared in both the head and neck tissue and pleural tumor, suggesting that the latter was a metastasis of primary HNSCC. Panel testing also showed high TMB in both tumors. Subsequently, the patient was treated with nivolumab, which resulted in significant tumor shrinkage and long-term survival of >32 months. In this case, genomic information obtained by panel testing was effective for both making an accurate diagnosis and selecting optimal pharmacotherapy [25].

### 3.6. Association between PD-L1 Expression and ICI response

To validate the other predictors, we studied 27 patients with R/M HNSCC treated with ICI at TMDU with a PD-L1 CPS (Table 2). There were 13 patients in the 1 ≤ CPS < 20 group and 14 in the CPS ≥ 20 group, and there were no CPS-negative (<1) cases. ORR was equivalent in two patients (14%) in the 1 ≤ CPS < 20 group and two patients (15%) in the CPS ≥ 20 group. Furthermore, the median PFS was 1.7 months (95% CI, 0.6–2.7) in the 1 ≤ CPS < 20 group and 2.3 months (95% CI, 1.0–5.5) in the CPS ≥ 20 group with no significant difference (*p* = 0.2, Figure 4A).

### 3.7. Association between Specific Genetic Mutation and ICI Response

In addition, the relationship between genetic mutations and the duration of ICI response was analyzed. Of the 34 SCC patients who underwent genomic testing, 32 were examined, excluding one who received chemotherapy with ICI and could not be evaluated due to interruption of the first round of treatment. Multivariate analysis was performed on the following frequent genetic variants: *TP53*, *NOTCH1*, *CDKN2A*, *PIK3CA*, *TERT promoter*, *CCND1,* and *FGF3*, *4*, *19* (Figure 4B). Since the co-amplification of the 11q13 region in the locus of *CCND1* and *FGF3*, *4*, *19* was detected, they were analyzed in the same group as *CCND1*. This amplification group (*n* = 6) showed a significantly lower ICI effect than the wild-type group (*n* = 26) (HR = 5.02 [95% CI, 1.56 to 16.2; *p* = 0.007). The best response to ICI therapy was PD in all cases in the *CCND1* and *FGF3*, *4*, *19* amplification groups. The median TMB was 5 (range, 5–9). Furthermore, the *CCND1* and *FGF3*, *4*, *19* amplification groups had significantly shorter median PFS (1.4 months, *p* = 0.0005) than their wild-type counterparts (median PFS: 2.6 months). The *PIK3CA* mutation group (*n* = 6) exhibited a lower ICI effect than the wild-type group (*n* = 26) (HR = 3.44, 95% CI, 1.11 to 10.6, *p* = 0.03), but no significant difference was observed in PFS (*p* = 0.2). No other mutations correlated with a lack of benefit in the response. These results suggest that *CCND1* may be a useful candidate gene for predicting the poor prognosis of ICI treatment. 

## 4. Discussion

This was a retrospective study of CGP, performed as a part of the standard of care for patients with HNC. Information on cancer gene panel tests conducted at all designated cancer care hospitals in Japan was integrated into the C-CAT database at the National Cancer Center, Japan, and case data for each cancer from October 2021 could be utilized. To the best of our knowledge, this is the first report in Japan utilizing the C-CAT database for HNC. Here, we analyzed treatment after oncogene profiling of 1119 HNC cases. A large difference was observed between the percentage of patients with actionable gene aberrations (23%) and those receiving targeted therapy (5%). A previous U.S. prospective cohort study found that only 11% of advanced cancer patients who underwent MSK-IMPACT testing were subsequently enrolled in genetically matched clinical trials [26]. The present results for HNCs are even less probable than theirs are. Targeted therapies include treatment for *NTRK* fusion and *ERBB2* amplification. The treatment of *NTRK* fusion-positive cancers with TRK inhibitors, such as entrectinib or larotrectinib, is associated with high response rates (>75%) and has been approved for treating advanced solid tumors [27]. *HER2*-overexpressing cancers have been treated with anti-HER2 trastuzumab and recently approved for HER2-positive salivary gland cancer [28]. 

Therefore, we examined the clinical utility of oncogene panel testing in 34 SCC and 20 non-SCC cases in an institution-based study. *TP53* mutations in cancer suppressor genes and *TERT* promoter mutations encoding telomerase have been detected in many cases of SCC compared to non-SCC. Consistent with a previous report [29], SCC cases showed *CDKN2A* inactivating mutations or deletions, *CCND1* amplification related to the cell cycle, *PIK3CA* activating mutations in the *PI3K* pathway, *MYC* amplification of oncogenes, *EGFR* amplification of receptor tyrosine kinases, and *NOTCH1* inactivating mutations encoding transmembrane receptors. In contrast, *CDKN2A/B* deletion, *MTAP* deletion, and *ERBB2* amplification were more common in the non-SCC cases. This study confirms the valuable genetic differences between histological HNC types, especially because genetic data on SCC are limited. Somatic mutations in tumor cells are expressed as the number of mutations per million base pairs (Muts/Mb), and the median TMB has been reported to be 5 Muts/Mb for HNSCC and 4 Muts/Mb for salivary gland cancer [26]. In this study, the median TMB was 5 Muts/Mb (0–34) in SCC cases and 3 Muts/Mb (0–11) in non-SCC cases, which is consistent with the results of previous reports. 

Actionable mutations were found in 88% of SCC and 74% of non-SCC patients, but only 9% of SCC and 16% of non-SCC patients received targeted drugs for their genetic aberrations. Genetically matched clinical trials or drugs are not available for the majority of patients. The low access rate to drugs, despite the detection of actionable gene mutations in many cases, may be due to drug indication and participation limitations in clinical trials (factors such as systemic conditions, medical history, and timing of trial openings). Analysis of both C-CAT and our data revealed that the development of specific gene-targeted therapy is slow in SCC. ICIs are currently recommended for a limited number of patients with high TMB. In contrast, *ERBB2* and *NTRK* mutations are considered druggable in non-SCC cases. The accumulation of tissue-specific gene analysis data is expected to lead to the development of new targeted therapies in the future.

The introduction of ICI therapy has revolutionized the treatment of HNSCC. However, only a minority of patients (<20%) responded to ICI monotherapy. Numerous biomarkers have been proposed; however, no robust markers have been proposed to predict the treatment efficacy. Predicting ICI response helps to avoid unnecessary toxicity and cost to patients and helps properly screen patients who may benefit from treatment. Typical clinical results of ICI therapy for HNSCC showed an ORR of 13% and a median PFS of 2.0 months for the nivolumab monotherapy group in the CheckMate 141 trial [3]. The pembrolizumab monotherapy group showed a 17% ORR and a median PFS of 2.3 months in the KEYNOTE-048 trial [6]. ICI therapy at our institution revealed comparable results of 18% ORR and a median PFS of 2.1 months. In this study, patients with high TMB (≥10) showed a long-term response, suggesting that TMB is a useful factor for predicting the response and prognosis of HNSCC. However, cases with TMB ≥ 10 in our institution comprised only 12% (4/33) of the SCC cases. TMB ≥ 20 was observed in 3% (1/33) of SCC cases, which is rarer than previously reported [23,26]. Additionally, in some cases in the TMB-low or intermediate groups, tumor shrinkage was observed initially, although the response could not be maintained, leading to tumor progression. This suggests that resistance may have been acquired during ICI therapy and that the majority of non-high TMB cases may benefit from ICI combined with molecularly targeted drugs that control a different pathway. We have previously reported that *MYC* is a candidate gene involved in acquiring ICI resistance [30]. However, we found no correlation between *MYC* expression and ICI treatment efficacy. Furthermore, the accumulation of cases is necessary to elucidate the mechanisms of ICI resistance.

There is no consensus on the association between HPV status and ICI response or TMB. In the KEYNOTE-12 study, HPV-positive HNSCC patients exhibited improved ICI response compared to HPV-negative patients, but this has not been observed in other studies [31]. Some studies have shown that virus-associated cancers have lower TMB [21,32], whereas others have shown no significant correlation [33]. We could not evaluate the significance of the HPV status in this study because of the small number of HPV-positive cases. PD-L1 expression in HNSCC has been as high as 46–100% in previous reports, and all patients in this study were CPS-positive (≥1). Our study showed no significant difference in ORR or median PFS according to the CPS value in positive cases, which is consistent with previous reports. This result suggests that the CPS value alone is insufficient to predict the prognosis of ICI and must be considered comprehensively with other relevant factors.

In addition, multivariate analysis of gene mutations showed that *CCND1* amplification was significantly associated with decreased response and poor prognosis in response to ICI therapy. *CCND1* encodes cyclin D1, a binding partner of CDK4/6, which regulates the retinoblastoma (RB) protein activity and cell cycle progression. *CCND1* amplification was observed in 36% of HNSCC cases [29]. The present study suggests that *CCND1* amplification has great potential as a biomarker for predicting the poor prognosis of ICI therapy in HNSCC, independent of TMB values and PD-L1 expression rates. To date, two meta-analyses have examined *CCND1* amplification and the effects of ICI. An investigation of the relationship between *CCND1* amplification and ICI treatment response rate reported a significantly shorter OS, especially in urothelial carcinoma (HR = 3.6) and melanoma (HR = 1.6–2.5) [34,35]. As functional evidence supporting a role for *CCND1* in the ICI response, Zhang et al. showed that PD-L1 protein abundance fluctuates during cell cycle progression and that cyclin D-CDK4 negatively regulates PD-L1 protein stability [36]. *CCND1* amplification can predict sensitivity to CDK4/6 inhibitors such as abemaciclib, palbociclib, and ribociclib, although these agents have reported only limited efficacy as monotherapy in tumor types other than breast cancer [37,38]. The combination of a CDK4/6 inhibitor and PD-1 antibody significantly inhibited tumor cell growth and improved survival in carcinomatous mice [36]. Multiple reports have suggested a synergistic antitumor effect of CDK4/6 inhibitors and ICIs, and their clinical application as combination therapies is expected. In addition, as the 11q13 region is the locus of *CCND1* and a group of *FGF* family genes such as *FGF3*, *FGF4*, and *FGF19*, their co-amplification was detected. Although no treatment strategy for *CCND1* has been established, FGFR inhibitors acting on *FGFs* may also affect *CCND1*. Several previous reports have shown that treatment of breast cancer cell lines with FGFR inhibitors downregulates *CCND1* and *CCND2* expression and inhibits CCND/CDK4 activity, resulting in decreased pRB phosphorylation [39,40]. The efficacy of combination therapy with anti-PD1 antibody drugs and FGFR inhibitors for *FGFR* and *FGF* amplification is currently undergoing clinical trials and should be analyzed in the future. 

However, the detailed function of CCND1 in the PD-1/PD-L1 pathway was not elucidated in the present study. The mechanism of *CCND* amplification and poor prognosis in immunotherapy require further investigation. In addition, no genetic mutations associated with favorable responses were identified in this study. Cancer gene panel testing is currently indicated at the end of standard treatment for the rarest cancers; therefore, there are few testing cases in patients with long-term responses to ICI therapy, which may lead to bias. The prediction of response requires the accumulation of genomic data on ICI responders. Due to the small number of cases in this study, there were limitations in identifying biomarkers associated with treatment response and resistance. Therefore, a larger sample size is required in future studies.

## 5. Conclusions

In this study, we evaluated the clinical use of CGP for HNC and discussed the issues associated with targeted therapy. The relationship between treatment response and oncogene profiling data in ICI-treated HNSCC patients was examined, and long-term response was observed in TMB-high cases. In contrast, *CCND1* amplification is predominantly associated with reduced reactivity to ICI and a poor prognosis. Therefore, a comprehensive oncogene analysis may be a useful tool for establishing response biomarkers for ICI therapy.

## Figures and Tables

**Figure 1 cancers-14-03476-f001:**
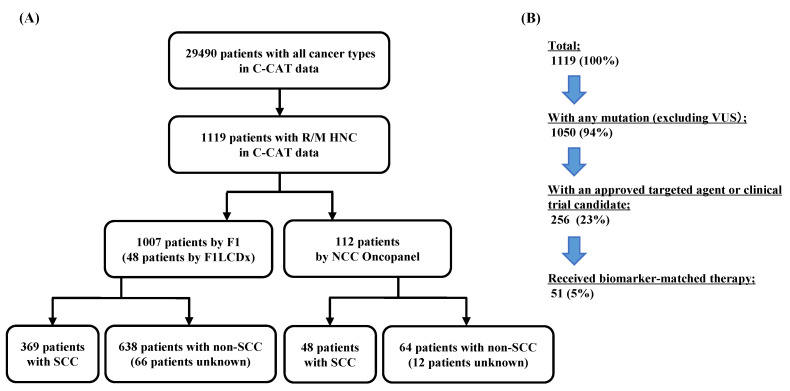
Total population and clinical utility of CGP in the C-CAT database. (**A**) C-CAT database included genomic information for the F1(F1CDx or F1LCDx) test and NCC Oncopanel test. Information of 29,490 patients was registered in C-CAT, with a total of 1119 patients diagnosed with R/M HNC. (**B**) Percentage of patients with access to treatment for comprehensive genomic profiles. VUS, a variant of uncertain significance.

**Figure 2 cancers-14-03476-f002:**
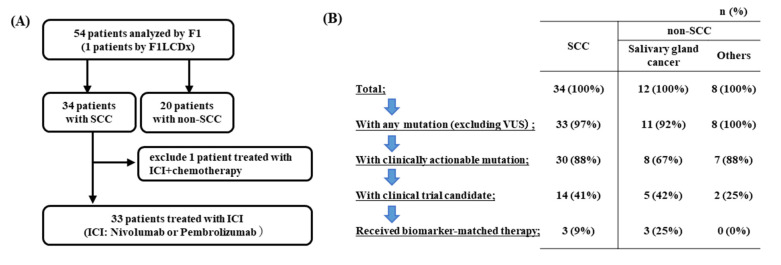
Patient population and sequencing results at our institution. (**A**) Total population in R/M HNC with F1CDx or F1LCDx testing performed at our institution. All SCC patients received ICI therapy; one patient, who received both ICI and chemotherapy, was excluded to evaluate treatment efficacy. (**B**) Percentage of patients with access to treatment for comprehensive genomic profiles by histology. (**C**) The top 30 most frequently detected genetic mutations in all R/M HNC patients. TMB is indicated at the top of the graph as high (red, ≥10 Mut/Mb), intermediate (yellow, 6–9 Mut/Mb), and low (blue, ≤5 Mut/Mb). The color coding on the graph indicates histological type, smoking history, and type of mutation. (**D**) Frequent genes and variant types by histological type. Color coding indicates mutation type.

**Figure 3 cancers-14-03476-f003:**
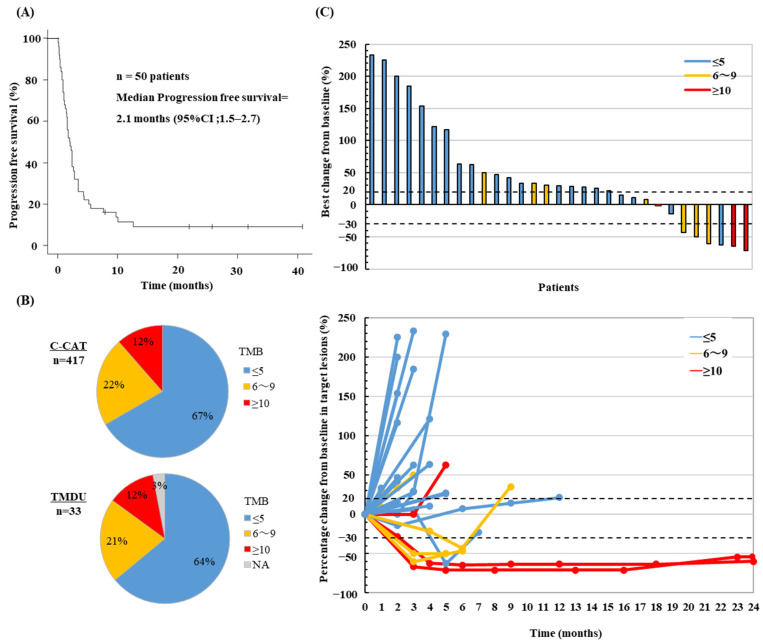
Outcomes of ICI therapy and association between TMB. (**A**) The Kaplan–Meier curves for progression-free survival among R/M HNSCC patients who have received ICI monotherapy to date in our department. (**B**) The percentage of TMB value in C-CAT and TMDU. (**C**) Response to ICI monotherapy in R/M HNSCC patients in the study. All 34 R/M HNSCC patients in this study had received ICI therapy. Of these, 1 patient who received ICI and chemotherapy, one patient who was not evaluable (NE) due to treatment interruption, and one patient with no detectable TMB value were excluded. For 31 patients, the waterfall plot (top) shows the best percent change from baseline in target lesions. Spider plot (bottom) showing objective response during ICI treatment. Color coding indicates TMB values; high (red, ≥10 Mut/Mb), medium (yellow, 6–9 Mut/Mb), and low (blue, ≤5 Mut/Mb). CI, confidence interval.

**Figure 4 cancers-14-03476-f004:**
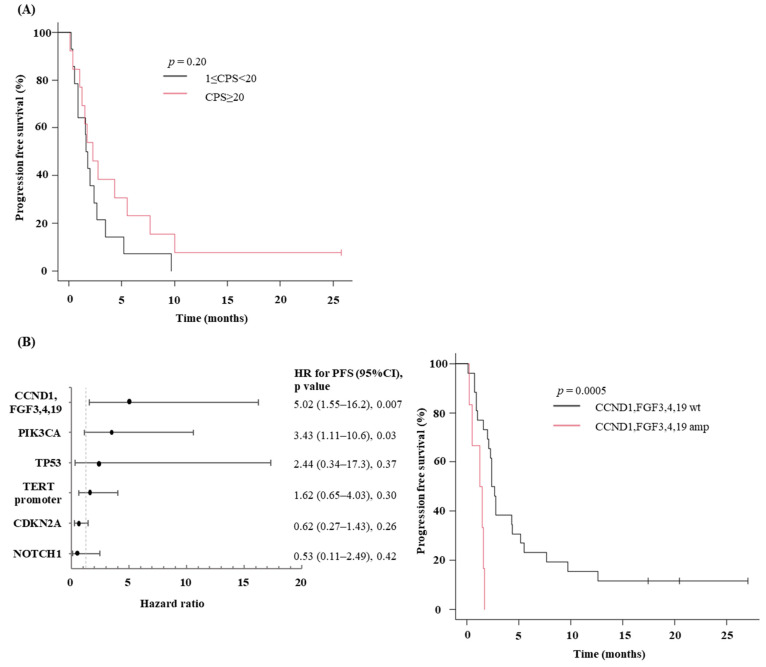
Outcomes of ICI therapy and association between other factors. (**A**) The Kaplan–Meier curves for progression-free survival by PD-L1 CPS value among R/M HNSCC patients who have received ICI monotherapy to date in our department. We analyzed 27 patients with measured PD L1 CPS. (**B**) Outcomes based on genetic alteration. Forest plots showing hazard ratios (HRs) with 95% CIs for progression-free survival (PFS). Kaplan–Meier curves for PFS in patients with CCND1 and FGF3, 4, 19 amplification or wild type group. Of the 34 R/M HNSCC patients in this study, 32 were analyzed, excluding one who received chemotherapy with ICI and one who could not be evaluated due to interruption of the first round of treatment. *p*-values are according to the log-rank test. HR, hazard ratio; amp, amplification; wt, wild-type.

**Table 1 cancers-14-03476-t001:** Clinical characteristics in our institution. NOS, not otherwise specified.

Characteristic	SCC	non-SCC
All patients	34 (100)	20 (100)
Age		
Median (range, y)	60 (42–81)	61 (15–74)
CorrectedGender, *n* (%)		
Male	21 (62)	9 (45)
Female	13 (38)	11 (55)
Primary site of disease, *n* (%)		
Oral cavity	25 (74)	-
Salivary glands	-	12 (60)
Nasal cavity and paranasal sinuses	4 (11)	2 (10)
Pharynx	2 (6)	-
Larynx	2 (6)	-
Parapharyngeal space	-	2 (10)
Unknown primary	1 (3)	1 (5)
Other	-	3 (15)
Histological classification, *n* (%)		
Squamous Cell Carcinoma	34 (100)	-
Adenoid cystic carcinoma	-	7 (35)
Salivary duct carcinoma	-	3 (15)
Carcinoma ex pleomorphic adenoma	-	2 (10)
Basal cell carcinoma	-	1 (5)
Esthesioneuroblastoma	-	1 (5)
Sarcoma	-	4 (20)
NOS	-	2 (10)
ECOG performance status, *n* (%)		
0	15 (44)	16 (80)
1	15 (44)	3 (15)
≥2	4 (12)	1 (5)
Smoking status, *n* (%)		
Never or <10 packs/year	12 (35)	10 (50)
Current or Former (≥10 packs/year)	22 (65)	10 (50)
Cancer staging, *n* (%)		
Stage I–Ⅲ	1 (3)	0 (0)
Stage IV	33 (97)	20 (100)
HPV, *n* (%)		
Negative	33 (97)	19 (95)
Positive	1 (3)	1 (5)

**Table 2 cancers-14-03476-t002:** Responses to ICI in patients with SCC.

Characteristic	*n* (%)	Best Response					ORR	*p*-Value	DCR	*p*-Value
		CR	PR	SD	PD	NE				
All patients	33 (100)	0 (0)	6 (18)	6 (18)	20 (60)	1 (3)	6 (18)		12 (36)	
TMB	33 (100)									
Low (≤5)	21 (64)	0 (0)	1 (5)	4 (19)	16 (76)	0 (0)	1/21 (5)	Ref	5/21 (24)	Ref
Intermediate (6–9)	7 (21)	0 (0)	3 (43)	0 (14)	4 (43)	0 (0)	3/7 (43)	0.03	3/7 (44)	0.5
High (≥10)	4 (12)	0 (0)	2 (50)	1 (33)	0 (0)	1 (1)	2/4 (50)	0.04	3/4 (75)	0.1
Not detected	1 (3)	-	-		-	-	-	-	-	-
PD-L1	27 (100)									
CPS < 1	0 (0)	0 (0)	0 (0)	0 (0)	0 (0)	0 (0)	0 (0)	NA	0 (0)	-
1 ≤ CPS < 20	14 (52)	0 (0)	2 (15)	0 (0)	10 (77)	1 (8)	2/14 (14)	NA	2/14 (14)	Ref
CPS ≥ 20	13 (48)	0 (0)	2 (14)	3 (21)	8 (57)	0 (0)	2/13 (15)	NA	5/13 (38)	0.2

**Table 3 cancers-14-03476-t003:** Clinical characteristics of patients with TMB-high.

Case No.	Age (yr)/Gender	Primary Site	Tumor Mutational Burden	Immune Checkpoint Inhibitor	Best Response
1	70/F	Buccal mucosa	13 mut/Mb	Pembrolizumab	PR
2	52/M	Maxillary sinus	34 mut/Mb	Nivolumab	PR
3	61/F	Buccal mucosa	10 mut/Mb	Pembrolizumab	SD
4	76/M	Unknown primary	14 mut/Mb	Nivolumab	NE

## Data Availability

Not applicable.

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
