# Peer review of "Comprehensive Genomic Profiling Reveals Clinical Associations in Response to Immune Therapy in Head and Neck Cancer"

_cancers, 2022, doi:10.3390/cancers14143476_

Round 1

Reviewer 1 Report

I think this paper needs re-focusing as it is looking at HNSCC and also other cancers of the head and neck (primarily salivary), which are two very distinct disease groupings with little overlap.  Points about targeted therapy are largely related to MASC and HER-2 therapy in non-squamous histologies and talking about this population doesn't add to conclusions regarding immunotherapy.

Based on the title the focus appears to be immunotherapy, and as such I would suggest focusing only on HNSCC and removing parts about non-HNSCC or discussing them in a separate paper.  This paper overall adds a little data to the CCND1 story, which is helpful, but again the patient numbers are unfortunately quite limited.   It also adds some real world data regarding response differences in PDL1 high and intermediate.

I am unsure what "actionable mutations were foudn in 85% and 74% of non-scc patients" refers to in the discussion as that would be a very unusual number of patients with approved targeted treatments unless that is referring to theoretical drugs that could potentially be given to target a mutation.

Author Response

Reviewer #1

I think this paper needs re-focusing as it is looking at HNSCC and also other cancers of the head and neck (primarily salivary), which are two very distinct disease groupings with little overlap.  Points about targeted therapy are largely related to MASC and HER-2 therapy in non-squamous histologies and talking about this population doesn't add to conclusions regarding immunotherapy.

Based on the title the focus appears to be immunotherapy, and as such I would suggest focusing only on HNSCC and removing parts about non-HNSCC or discussing them in a separate paper. 

Response:

Thank you for raising these points. As mentioned in introduction, we evaluated (1) the real-world data of genetic and clinical utility in R/M HNC by conducting the nationwide database and institution-base analysis, and further (2) the association between ICI response in HNSCC and multiple candidate predictors, including TMB and PD-L1 expression rates, and gene alterations using CGP. In this study, we have conducted comprehensive analysis of head and neck cancers including non-SCC such as salivary tumors using CGP. Our analysis from C-CAT database showed non-SCC cases included patients who received biomarker-matched treatment for NTRK3 fusion or ERBB2 amplification. This is the first report to show the clinical utility of CGP in head and neck cancers in Japan and we believe this result is quite informative to the field.

This paper overall adds a little data to the CCND1 story, which is helpful, but again the patient numbers are unfortunately quite limited. 

Response:

We agree that the numbers of patients are limited. However, some meta-analyses already showed CCND1 alteration was associated with the effect of ICIs, especially in urothelial carcinoma and melanoma. Also, in vitro and in vivo, PD-L1 protein was shown to fluctuate in multiple human cancer cell lines during cell cycle progression and Cyclin D-CDK4 negatively regulates PD-L1 protein stability. We feel the mechanism of CCND1 amplification and poor prognosis in immunotherapy requires further investigation, however, together with our data and previous reports, we suggest our real world data in HNC would be helpful to identify biomarkers to predict the effect of ICIs.

It also adds some real world data regarding response differences in PDL1 high and intermediate.

Response:

We have shown that ORR was equivalent in PDL1 intermediate and high group and furthermore, the median PFS was not significantly different between PDL1 intermediate and high group (p=0.2, Fig 4A). We respectfully feel that the inclusion of the real world data of the patients with PDL1 high and intermediate does not add further information.

I am unsure what "actionable mutations were foudn in 85% and 74% of non-scc patients" refers to in the discussion as that would be a very unusual number of patients with approved targeted treatments unless that is referring to theoretical drugs that could potentially be given to target a mutation.

Response:

We have revised the statement regarding actionability in Methods. Actionable mutation is defined as a genomic alteration that satisfies the following conditions: gene alteration is characterized as a target of molecular targeted therapy, and a drug is available for human use either as an antibody or as a small molecule compound with a nanomolar scale IC50 concentration. Actionable gene aberrations are supposed to confer sensitivity to either an approved or an experimental targeted agent that is currently in clinical trials. We have revised the percentage of access to treatment in Fig. 2D.

Reviewer 2 Report

This is a study on comprehensive genomic profiling and its implications in head and neck cancer. The authors included CGP from a national database as well as from their own institution, focusing in the latter half of the paper on responses to immune checkpoint blockade. Overall, the paper is interesting as a real-world study of how CGP is used in clinical practice. Figure 3 is particularly great. However, the data and flow of the study are not cohesive and at times it is difficult to know what the paper is about. Is this about genomic profiling? Targeted therapy? Immunotherapy resistance? More cohesive writing to tie these themes together better would improve the manuscript.

Major concerns:
It is unclear why salivary tumors were included from the national database. What does this add to the paper? Further, salivary tumors are not mentioned in the title or abstract. As the authors know, salivary tumors are very different from head and neck squamous cell carcinoma. Would consider removing the data on salivary tumors, or alternatively, integrating those data better by mentioning salivary cancer in the abstract/title. 

Minor concerns:

1) Section 3.2 of the Results is redundant, since the information is all in Table 1. Consider making this paragraph more concise. 

2) Consider changing "former smoker" in  Figure 2 to "current or former smoker" since I assume some patients have not quit smoking?

Author Response

Reviewer #2

This is a study on comprehensive genomic profiling and its implications in head and neck cancer. The authors included CGP from a national database as well as from their own institution, focusing in the latter half of the paper on responses to immune checkpoint blockade. Overall, the paper is interesting as a real-world study of how CGP is used in clinical practice. Figure 3 is particularly great. However, the data and flow of the study are not cohesive and at times it is difficult to know what the paper is about. Is this about genomic profiling? Targeted therapy? Immunotherapy resistance? More cohesive writing to tie these themes together better would improve the manuscript.

Response:

We have revised the abstract to clarify the aim of this manuscript. As mentioned in introduction, we evaluated (1) the real-world data of genetic and clinical utility in R/M HNC by conducting the nationwide database and institution-base analysis, and further (2) the association between ICI response in HNSCC and multiple candidate predictors, including TMB and PD-L1 expression rates, and gene alterations using CGP. We believe our approach is reasonable and helpful to understand the heterogeneity of HNC.

Major concerns:

It is unclear why salivary tumors were included from the national database. What does this add to the paper? Further, salivary tumors are not mentioned in the title or abstract. As the authors know, salivary tumors are very different from head and neck squamous cell carcinoma. Would consider removing the data on salivary tumors, or alternatively, integrating those data better by mentioning salivary cancer in the abstract/title.

Response:

Thank you for raising these points. In this study, we have conducted comprehensive analysis of head and neck cancers including non-SCC such as salivary tumors using CGP. Our analysis from C-CAT database showed non-SCC cases included patients who received biomarker-matched treatment for NTRK3 fusion or ERBB2 amplification. This is the first report to show the clinical utility of CGP in head and neck cancers in Japan and we believe this result is quite informative to the field. We have revised the percentage of access to treatment in Fig. 2D and mentioned salivary grand cancers in the abstract.

Minor concerns:

1) Section 3.2 of the Results is redundant, since the information is all in Table 1. Consider making this paragraph more concise.

Response:

We have deleted some description from this paragraph and made it more concise.

2) Consider changing "former smoker" in Figure 2 to "current or former smoker" since I assume some patients have not quit smoking?

Response:

Thank you for this suggestion. Corrected.

Reviewer 3 Report

This study has an interesting approach, and the accuracy of the models chosen for the clinical evaluation, the data collection, and the manner of statistical analysis are to be appreciated.

The study analyzes the clinical utility of comprehensive genomic profiling (CGP) testing to identify predictive biomarkers that respond to Immune checkpoint inhibitors (ICIs) in recurrent/metastatic head and neck cancers (R / M HNC). By using the nationwide genetic clinical database established by the Center for Cancer Genomics and Advanced Therapeutics, a complex statistical analysis was performed that generated valuable information about the response to ICI in squamous cell carcinoma (SCC) cases.

CGP using NGS has been used in clinical practice to select effective targeted therapies. An alternative biomarker is tumor mutation load (TMB), as measured by next-generation sequencing (NGS). Thus, were evaluated the top 30 most frequent genetic mutations in HNC patients with pharmacological intervention. Data obtained suggest that as matched-targeted therapy patients with TMB-high SCC were subjected to immunotherapy, whereas patients with non-SCC were treated with molecularly targeted drugs against ERBB2 mutations.

The study is valuable for the genomic information obtained that was effective both in establishing an accurate diagnosis and in selecting the optimal pharmacotherapy. Analysis of the relationship between genetic mutations and the duration of the ICI response suggests that CCND1 may be a useful biomarker candidate for predicting the poor prognosis of ICI therapy in HNSCC.

This article is well designed, uses recent literature data, and can be interesting for readers.

There are some typographical and grammatical errors in the manuscript that I have highlighted. Overall, I consider the article could be a useful contribution to the journal. I recommend the manuscript for publishing.

Author Response

Reviewer #3

This study has an interesting approach, and the accuracy of the models chosen for the clinical evaluation, the data collection, and the manner of statistical analysis are to be appreciated.

The study analyzes the clinical utility of comprehensive genomic profiling (CGP) testing to identify predictive biomarkers that respond to Immune checkpoint inhibitors (ICIs) in recurrent/metastatic head and neck cancers (R / M HNC). By using the nationwide genetic clinical database established by the Center for Cancer Genomics and Advanced Therapeutics, a complex statistical analysis was performed that generated valuable information about the response to ICI in squamous cell carcinoma (SCC) cases.

CGP using NGS has been used in clinical practice to select effective targeted therapies. An alternative biomarker is tumor mutation load (TMB), as measured by next-generation sequencing (NGS). Thus, were evaluated the top 30 most frequent genetic mutations in HNC patients with pharmacological intervention. Data obtained suggest that as matched-targeted therapy patients with TMB-high SCC were subjected to immunotherapy, whereas patients with non-SCC were treated with molecularly targeted drugs against ERBB2 mutations.

The study is valuable for the genomic information obtained that was effective both in establishing an accurate diagnosis and in selecting the optimal pharmacotherapy. Analysis of the relationship between genetic mutations and the duration of the ICI response suggests that CCND1 may be a useful biomarker candidate for predicting the poor prognosis of ICI therapy in HNSCC.

This article is well designed, uses recent literature data, and can be interesting for readers.

There are some typographical and grammatical errors in the manuscript that I have highlighted. Overall, I consider the article could be a useful contribution to the journal. I recommend the manuscript for publishing.

Response:

Thank you for your positive comment. We have accordingly corrected the typographical and grammatical errors.

Reviewer 4 Report

This is an interesting study about comprehensive genomic profiling and clinical associations in response to immune therapy in head and neck cancer.

The paper is well written. However, some issues remain.

All the acronyms should be explained at their first appearance in the abstract (i.e., TMB).

I think that the authors should analyze patients according to two subgroups: squamous cell carcinomas and salivary tumors. Other rare tumors should be excluded to reduce biases.

Moreover, TMB can be different in HPV-positive tumors compared to HPV-negative ones. Therefore, analyses must be taken into account HPV status.

Please add p values in table 2.

Since different mutations are present in HPV-positive and HPV-negative carcinomas, association analyses between specific genetic mutation and ICI response must be performed separately in the two subgroups (HPV+ versus HPV-).

Author Response

Reviewer #4

This is an interesting study about comprehensive genomic profiling and clinical associations in response to immune therapy in head and neck cancer.

The paper is well written. However, some issues remain.

All the acronyms should be explained at their first appearance in the abstract (i.e., TMB).

Response:

We have added the explanation regarding TMB in the abstract.

I think that the authors should analyze patients according to two subgroups: squamous cell carcinomas and salivary tumors. Other rare tumors should be excluded to reduce biases.

Response:

Thank you for this suggestion. We have revised the percentage of access to treatment in Fig. 2D to avoid the biases and mentioned salivary grand cancers in the abstract.

Moreover, TMB can be different in HPV-positive tumors compared to HPV-negative ones. Therefore, analyses must be taken into account HPV status.

Response:

We agree with this important comment. We have accordingly revised and added HPV status in Fig. 2B. There is only one HPV-positive case in SCC and we could not evaluate the significance of HPV status in this study because of the small number of HPV-positive cases.  

Please add p values in table 2.

Response:

We have added p-value of ORR, DCR in Table 2.

Since different mutations are present in HPV-positive and HPV-negative carcinomas, association analyses between specific genetic mutation and ICI response must be performed separately in the two subgroups (HPV+ versus HPV-).

Response:

We thank the reviewer for raising this good point. There is no consensus regarding the association between HPV status and ICI response or TMB. In the KEYNOTE-12 study, HPV-positive HNSCC had improved ICI response compared to negative patients, but this has not been observed in other studies1. Some studies have shown that virus-associated cancers have lower TMB2,3, while others have shown no significant correlation4. We could not evaluate the significance of HPV status in this study because of the small number of HPV-positive cases. We have added this statement in the discussion.

References

  1. Chow LQM, Haddad R, Gupta S, et al. Antitumor Activity of Pembrolizumab in Biomarker-Unselected Patients With Recurrent and/or Metastatic Head and Neck Squamous Cell Carcinoma: Results From the Phase Ib KEYNOTE-012 Expansion Cohort. J Clin Oncol. 2016;34: 3838-3845.
  2. Hanna GJ, Lizotte P, Cavanaugh M, et al. Frameshift events predict anti-PD-1/L1 response in head and neck cancer. JCI Insight. 2018;3.
  3. Zhang L, Li B, Peng Y, et al. The prognostic value of TMB and the relationship between TMB and immune infiltration in head and neck squamous cell carcinoma: A gene expression-based study. Oral Oncol. 2020;110: 104943.
  4. Kumar G, South AP, Curry JM, et al. Multimodal genomic markers predict immunotherapy response in the head and neck squamous cell carcinoma. bioRxiv. 2021.

Round 2

Reviewer 1 Report

I think the paper's edits have improved it.  I understand the desire to provide a broad overview of CGP in HNSCC and head and neck cancer more broadly.  I still find the flow somewhat confusing and would recommend making non-HNSCC and HNSCC clearly delineated and separated where possible as these are totally different entities with one being predominantly one histology and the other entity encompassing many histologies.  HNC to me is a confusing acronym and I would find something such as non-squamous H&N cancer or something to that effect a better signifier

Author Response

Thank you for your positive comment and understanding our approach to this study. We agree that there are many histological types in non-HNSCC and classified them in Table 1, which could make it easier for readers to understand the difference between SCC and non-SCC in regard to primary site and histology. We have added the explanation to HNC and R/M HNC at the first appearance in introduction.

Reviewer 4 Report

Thank you for improving the manuscript.

Author Response

Thank you for your positive feedback.